# Exploring the High Frequencies AC Conductivity Response in Disordered Materials by Using the Damped Harmonic Oscillator

Christos Tsonos

Department of Physics, University of Thessaly, 3rd Km O.N.R. Lamia-Athens, 35100 Lamia, Greece; christostsonos@uth.gr

**Abstract:** The AC conductivity response of disordered materials follows a universal power law of the form $\sigma'(\omega) \propto \omega^n$ at the low frequency regime, with the power exponent values in the range $0 < n < 1$. At the high frequency regime, in many experimental data of different disordered materials, superlinear values of the power exponent $n$ were observed. The observed superlinear values of the power exponent are usually within $1 < n < 2$, but in some cases values $n > 2$ were detected. The present work is based on the definitions of electromagnetic theory as well as the Havriliak–Negami equation and the damped harmonic oscillator equation, which are widely used for the description of dielectric relaxation mechanisms and vibration modes in the THz frequency region, respectively. This work focuses mainly on investigating the parameters that affect the power exponent and the range of possible $n$ values.

**Keywords:** AC conductivity; vibration modes; disordered materials; composite materials; simulation

## 1. Introduction

The alternating current (AC) conductivity, as described by the real part, $\sigma'(\omega)$, of the complex electrical conductivity $\sigma^*(\omega) = \sigma'(\omega) + j\sigma''(\omega)$, presents similar behavior in various kinds of disordered conductive materials such as polymers, composites, semiconductors, ionic glasses, and ceramics. AC conductivity $\sigma'(\omega)$ has a universal form, given by the relation $\sigma'(\omega) \propto \omega^n$, where $\omega$ denotes the angular frequency and $n$ is a power exponent which, at the low frequency regime, takes values $0 < n < 1$ [1,2]. The previous relation is known as the universal dynamic response (UDR). Several theoretical approaches have been developed in order to interpret this behavior [3–9]. It has been also reported in various theoretical approaches that the value of exponent $n$ has a unique value close to 0.7 [10–13]. Let us note that values of $n > 1$ have also been reported, for different types of disordered materials subjected to dielectric measurements at the low frequency regime, below 10 MHz [14–23].

In a previous work it was shown that cases where the power exponent $n$ gets values in the range $0 < n < 1$ could be directly related to the contribution of mobile charges if, in the frequency spectrum under study, they contribute only the direct current (DC) conduction and the slowest polarization mechanism, due to the charge motions within sort length scales [24]. Apparent $n$ values in the range $1 \le n \le 2$, for a relatively narrow frequency range, should be attributed to an additional molecular dipolar relaxation contribution that takes place at higher frequencies [24]. Between two well-separated dielectric mechanisms, with a clear shallow minimum in the imaginary part of complex permittivity $\varepsilon^*$, the apparent power exponent $n$ can take values higher than 1 for a relatively extended frequency range [24].

At the high frequency regime and especially in the GHz to THz frequency region, in many experimental data of different disordered materials, values of power exponent $n$ equal to or higher than 1 were detected [25–29]. A nearly linear increase of $\sigma'$ at higher

frequencies, corresponding to near constant losses (NCL), has been also reported [11,30]. However, measurements extending up to high frequencies or to low temperatures reveal a superlinear behavior of the power exponent, $n > 1$ [26,27]. Usually, the observed superlinear values of the power exponent are within $1 < n < 2$, but it should be mentioned that in glycerol and for lower temperatures the increase of the imaginary part of the complex dielectric constant, $\varepsilon''$, follows a power law $\omega^3$ at the 400–900 GHz frequency region [31], which implies a value of power exponent of $n = 4$. This is one of the higher values of power exponent of AC conductivity which has been detected in the respective frequency range to the best of our knowledge. On the other hand, the higher frequency range is an interesting region for the study of promising materials for microwave applications [32,33]. The combination of different compounds which have excellent microwave properties leads to new composite materials which have earned great technological interest in recent years, while the addition of a second phase can significantly improve the electronic properties of the resulting composite material [34,35].

The present work is based on the definitions of electromagnetic theory as well as the Havriliak–Negami (HN) equation and the damped harmonic oscillator (DHO) equation, which are widely used for the description of dielectric relaxation mechanisms and vibration modes in the THz frequency region, respectively. The purpose of the present work is to investigate and to discuss the frequency dependent AC conductivity at higher frequency regimes—up to lower frequencies of far infrared (FIR) spectra—in disordered and composite materials. The power exponent, $n$, is a crucial parameter that characterizes the AC conductivity response of the materials. This work focuses mainly on investigating the parameters that affect the power exponent and the range of possible $n$ values. This study has been based on fitting the data and performing calculations and simulations in polymeric and glassy systems.

## 2. Theoretical Definitions and Relations

The complex electrical conductivity, $\sigma^*(\omega)$, is related to the complex dielectric constant, $\varepsilon^*(\omega) = \varepsilon'(\omega) - j\varepsilon''(\omega)$, through the relation:

$$\sigma^*(\omega) = j\omega\varepsilon_o\varepsilon^*(\omega) \tag{1}$$

If the direct current (DC) conductivity, $\sigma_o$, is subtracted from $\varepsilon^*(\omega)$ then Equation (1) becomes:

$$\sigma^*(\omega) = \sigma_o + j\omega\varepsilon_o\varepsilon_d^*(\omega) \tag{2}$$

where $\varepsilon_d^*(\omega)$ represents the complex dielectric constant caused only from dielectric losses mechanisms.

For the description of dielectric losses mechanisms, the HN equation is used [36]:

$$\varepsilon_d^*(\omega) = \varepsilon_\infty + \frac{\Delta\varepsilon}{\left(1 + (j\omega\tau)^\alpha\right)^\beta} \tag{3}$$

where $\varepsilon_\infty$ is the dielectric constant at the high frequency limit, $\Delta\varepsilon$ is the dielectric strength and $\tau$ is a characteristic relaxation time related to a characteristic frequency $\omega_{HN}$ via the relation $\omega_{HN}\tau = 1$. The characteristic frequency $\omega_{HN}$ is connected to the loss peak frequency $\omega_{max}$ via the relation $\omega_{max} = A\omega_{HN}$ where A is a constant depending from shape parameters $\alpha$ and $\beta$ [36].

Consider a conductive disordered material which is characterized, in the frequency spectra under study, only by the contribution of DC conductivity, $\sigma_o$, and a dielectric dispersion. In this case, the real part of the complex conductivity as extracted from Equations (2) and (3), is given by the relation:

$$\sigma'(\omega) = \sigma_o + \varepsilon_o\omega\varepsilon_d''(\omega) \tag{4}$$

In the case $\sigma_o \geq \varepsilon_o \omega \varepsilon_d''(\omega)$ at frequencies $\omega \leq \omega_{HN}$, then at $\omega > \omega_{HN}$ as $\omega$ increases Equation (5) is written [24]:

$$\sigma'(\omega) \cong \sigma_o + \varepsilon_o \omega_{HN}^{\alpha\beta} \Delta\varepsilon \sin(\alpha\beta\pi/2)\omega^{1-\alpha\beta} \tag{5}$$

Equation (5) expresses what usually is observed and describes the majority of AC conductivity response in disordered materials. The power exponent $n = 1\text{-}\alpha\beta$ takes values in the range $0 < n < 1$. DC charge mobility, in disordered materials, always results in a polarization process, which is due to the charge motions within short length scales according to the Random Barrier model [10]. This polarization mechanism obeys the Barton, Nakajima and Namikawa (BNN) relation and it is usually masked by DC conductivity effects or it appears as a shoulder [37]. Thus, the faster components of the polarization mechanism form the value of high frequencies slope of AC conductivity and are related to the short-range charges motion.

In the case $\sigma_o < \varepsilon_o \omega \varepsilon_d''(\omega)$ at frequencies $\omega \leq \omega_{HN}$, for several orders of frequency magnitude, then Equation (4) in the frequency range $\omega \leq \omega_{HN}$ is written as [24]:

$$\sigma'(\omega) \cong \sigma_o + \varepsilon_o \omega_{HN}^{-\alpha} \Delta\varepsilon\beta \sin(\alpha\pi/2)\omega^{1+\alpha} \tag{6}$$

This situation could take place only if the HN dispersion characterizes molecular dipolar relaxations [24]. The power exponent of AC conductivity is equal to $1 + \alpha$ and it should take values in the range $1 < n \leq 2$, for a restricted frequency range. Thus, in the previous situation the slower components of the molecular dipolar dispersion form the value of the high frequencies slope of AC conductivity. Also, it should be mentioned that in the case of the existence of two dispersions, the suitable coupling of HN parameter values can lead to an apparent NCL in a relatively extended frequency range, giving rise to a power exponent value close to 1 [24].

The DHO equation is widely used to describe the dielectric response of the higher frequency vibration modes. In this case, the complex dielectric constant is given by the following equation [38]:

$$\varepsilon_V^*(\omega) = \varepsilon_\infty + \frac{A}{\omega_0^2 - \omega^2 + j\omega\gamma} \tag{7}$$

where $\varepsilon_\infty$ is the high frequency dielectric constant, $\omega_o$ is the resonance frequency, $A$ is the oscillator strength and $\gamma$ is the damping constant.

## 3. Results and Discussion

Usually, at the lower frequency regime, the main contributions to the total losses are the conduction losses due to the free charge carriers and the rotational type losses due to dipoles or dipole-like processes. At higher frequencies, in the range of few THz, the contributions from vibration modes usually take place [38–44]. These modes are expected to affect the lower frequency regime [26,29]. The DHO equation is widely used to describe the dielectric response of vibration modes of various materials [38,41–44]. The imaginary part of the DHO equation is given by the relation:

$$\varepsilon_V'' = \frac{\omega\gamma A}{(\omega_o^2 - \omega^2)^2 + \omega^2\gamma^2} \tag{8}$$

At $\omega << \omega_o$ the frequency dependence of dielectric losses is $\varepsilon_V'' \propto \omega$, and the AC conductivity behaves as $\sigma_V'(\omega) \propto \omega^2$, based on Equation (2). Therefore, the predominance of the vibration mode term $\varepsilon_o \omega \varepsilon_V''(\omega)$ against the other low frequency contributions tends to form a slope close to 2 at the extra high frequency range of GHz, in $\log \sigma' - \log \omega$ plots. In order to determine the effect of vibrational modes in the AC conductivity response up to FIR frequency range, simulations have been made based on the following relation:

$$\sigma'(\omega) = k\sigma_o + l\varepsilon_o \omega \text{Im}[HN] + m\varepsilon_o \omega \text{Im}[DHO] \tag{9}$$

where $k$, $l$, $m$ = 0 or 1, while Im [$HN$] and Im [$DHO$] denote the imaginary part of Equations (3) and (7) respectively.

Figure 1 describes the AC conductivity response, where the DC conductivity, one dielectric relaxation at the low frequency regime, and one vibration mode were included (for details see Figure 1 caption). As shown in Figure 1, the high frequency slope remains constant at a value of $n = 1-\alpha\beta = 0.67$ for the curve (a) that corresponds to the case $k = l = 1$ and $m = 0$ where the vibration mode is absent. For the case $k = l = m = 1$, the dielectric effect from one vibrational mode is also included in Figure 1 (curve (b)). Typical values close to those of the literature were used for the parameters of Equation (9) [41–43]. Deviation from the value of $n = 0.67$ begins to occur for frequencies higher than 1 GHz in curve (b), for the particular parameter values. In what follows, the lower frequencies of FIR range, 300–600 GHz, was selected as the reference range for the calculation of the power exponent $n$ of AC conductivity. The slope at the GHz region gradually increases and reaches a value of 1.92 at the frequency range of 300–600 GHz, well below the resonance peak at 2.5 THz.

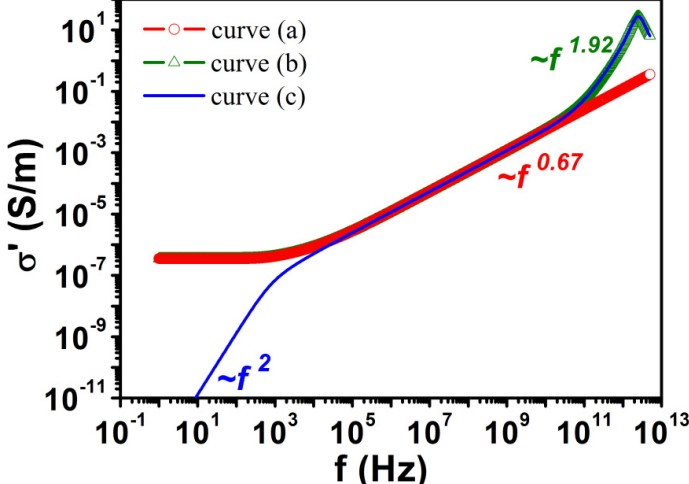

**Figure 1.** Simulation curves according to Equation (9) with parameter values: $\sigma_o = 3.48 \times 10^{-7}$ S/m, $\alpha = 1$, $\beta = 0.32$, $\Delta\varepsilon = 4.63$, $f_{HN} = \omega_{HN}/2\pi = 637$ (Hz), $\gamma/2\pi = 2 \times 10^{12}$ (Hz), $A/(2\pi)^2 = 1 \times 10^{24}$ (Hz$^2$) and $\omega_o/2\pi = 2.5 \times 10^{12}$ (Hz). (**a**) Simulation curve of Equation (16) with $k = l = 1$ and $m = 0$. (**b**) Simulation curve of (16) with k = l = m = 1. (**c**) Simulation curve of Equation (9) with k = 0 and l = m = 1.

Table 1 shows the power exponent of AC conductivity value at the frequency range 300–600 GHz, for various parameters values of Equation (7), according to Equation (9). The power exponent $n$ presents superlinear behavior with values ranging from 1.23 up to 2, as shown in the same table. For the same value of $\omega_o$, the increase of $A$ (or $\gamma$) leads to an increased value of power exponent in the respective frequency range. Also, for the same values of parameters $\gamma$ and $A$, as the characteristic frequency $\omega_o$ increases, the power exponent decreases in the respective frequency range. The simulation curves of Equation (9), which include the contribution of one vibration mode as described from the DHO equation, give superlinear values of the power exponent at the lower frequencies of the FIR range. The power exponent values depend on the correlation of the dynamic characteristics of the vibration modes. It should be noted here that, theoretically, the existence of one Debye relaxation or Cole–Davidson dispersion ($\alpha = 1$) at THz frequency region could have a similar effect on the power exponent in the GHz frequency range.

**Table 1.** The power exponent of AC conductivity at frequency range 300–600 GHz for various parameters values of Equation (8), according to Equation (9). The values of the rest parameters of Equation (9) were keeping constant as those of Figure 1.

| $\gamma/2\pi$ (THz) | $A/(2\pi)^2$ (THz$^2$) | $\omega_o/2\pi$ (THz) | $n$ |
|---|---|---|---|
| 2 | 3 | 2.5 | 2.00 |
| 2 | 1 | 2.5 | 1.92 |
| 2 | 3 | 5.0 | 1.46 |
| 1 | 3 | 5.0 | 1.23 |
| 1 | 1 | 2.5 | 1.82 |

Finally, it is useful to mention here a special case which usually characterizes the low temperature response: the absence of DC conductivity contribution which corresponds to the case $k = 0$ and $l = m = 1$ of Equation (9). In this case, there is a significant difference at the lower frequency range in $\log \sigma' - \log \omega$ plots as shown in Figure 1 (curve (c)). The slope in the lowest frequency range is shaped by the low frequency regime of HN mechanism. At this frequency region it is $\varepsilon''_d \propto \omega^\alpha$, and hence $\sigma'(\omega) \propto \omega^{1+\alpha}$ instead of the DC conductivity plateau. In our case $\alpha = 1$ and hence $\sigma'(\omega) \propto \omega^2$ (Figure 1, curve (c)).

The existence of an additional HN mechanism at the intermediate frequencies (MHz–GHz) of Figure 1, can lead to a differentiation of the frequency dependence of the AC conductivity. In this case, a remarkable effect on the slope of AC conductivity is expected. Figure 2 shows the broad-band AC conductivity spectra of poly(p-phenylenediamine) (PPDA) in which measurements in microwave (MW) and FIR frequencies are included [45]. In order to fit the data of Figure 2, Equation (9) is used with $k = m = 1$ and two HN terms. According to the best fit, all data from lowest up to highest frequencies are described very well as shown in Figure 2. In the same figure, a simulation curve without taking into account the intermediate HN mechanism is presented. It is obvious that the existence of an intermediate HN mechanism has a remarkable influence on the GHz frequency range as shown in Figure 2.

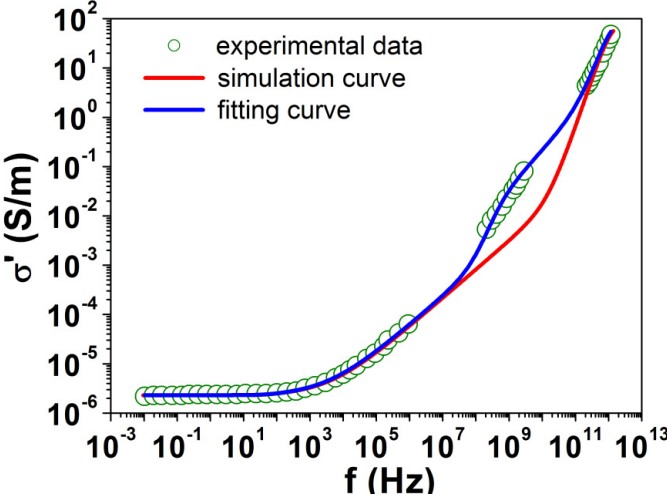

**Figure 2.** Experimental data of the AC conductivity as a function of frequency of poly(p-phenylenediamine) (PPDA) (Data were taken from literature [45]). The blue line is the best fitting according to Equation (9) with k = m = 1 and two HN terms, by keeping constant the DC conductivity value, $\sigma_o = 2.3 \times 10^{-6}$ S/m. The fitting parameters' values are: $\alpha_1 = 1$, $\beta_1 = 0.42$, $\Delta\varepsilon_1 = 114.5$, $f_{HN1} = \omega_{HN1}/2\pi = 49.0$ (Hz), $a_2 = 1$, $\beta_2 = 0.28$, $\Delta\varepsilon_2 = 2.1$, $f_{HN2} = \omega_{HN2}/2\pi = 410$ (MHz), $\gamma/2\pi = 2.0 \times 10^{12}$ (Hz), $A/(2\pi)^2 = 4.5 \times 10^{24}$ (Hz$^2$) and $\omega_o/2\pi = 3.9 \times 10^{12}$ (Hz). The red line is a simulation curve according to Equation (9) with the previous parameter values and without the second HN contribution.

Let us see now the influence of vibrational modes on the MW frequency range in glass material. Glass-forming systems are a special class of materials with specific characteristics in their dynamic response. In these materials a loss peak shows up at some THz that can be identified with the so-called boson peak. A variety of explanations of the boson peak has been proposed, such as the soft potential model [46,47], phonon localization models [48,49] and a model of coupled harmonic oscillators with a distribution of force constants [50]. Also, the occurrence of the boson peak has been modeled within the mode-coupling theory (MCT) [51]. Between the $\alpha$-peak at lower frequencies and the boson peak, obviously a minimum in $\varepsilon''$ must exist, as found in a variety of dielectric spectroscopy measurements. The experimental results indeed provide evidence for a fast process in this region, as the spectral response near the minimum cannot be explained assuming a simple superposition of $\alpha$-peak, including excess wing contribution, and boson peak. This frequency region was mainly stimulated by the MCT, which predicts that a fast process will lead to significant additional contributions in this minimum region [52–55]. In the frequency range ~1–300 GHz, the fast relaxation makes a noticeable contribution to the dynamic response of glass-forming materials [56]. Figure 3 shows the AC conductivity as a function of frequency of a lithium silicate glass at room temperature, which includes measurements in MW and FIR frequencies [29]. The slope in the MW range from 1–3 GHz is equal to 1.26. A sum of three terms, $\varepsilon_o \omega_i \text{Im}[DHO]_i$, were used for the best fitting of the FIR data, which include the contribution of three vibration modes. In the same figure the MW data after the subtraction of vibration modes contribution are also presented. The slope of AC conductivity, in free FIR contribution data, is equal to 1.15 at the same MW frequency range, a value which remains higher than unity. The only way to explain this superlinear value of the power exponent of AC conductivity is the contribution of a low frequency regime of a dielectric losses mechanism which present peak in $\varepsilon''$ at frequencies higher than 3 GHz. The existence of a well-defined shallow minimum in $\varepsilon''$ gives rise to the appearance of a superlinear value of $n$ at the lower GHz frequency range [24], without the FIR vibrational contribution. Therefore, the superlinear value of the power exponent of AC conductivity in free FIR contribution data of Figure 3 should be attributed to the influence of the fast relaxation.

Finally, it is important to point out that in some cases, depending on the DHO parameters correlation, it is possible to detect a power exponent of AC conductivity values higher than 2 below the resonance frequency $\omega_o$. As mentioned previously, at $\omega << \omega_o$ the frequency dependence of dielectric losses is $\varepsilon''_V \propto \omega$, and so the AC conductivity behaves as $\sigma'_V(\omega) \propto \omega^2$. For frequency regions just below $\omega_o$, in some cases the dielectric losses of vibration modes could have a frequency dependence $\varepsilon''_V \propto \omega^p$, where $p > 1$, and therefore the AC conductivity could behave as $\sigma'_V(\omega) \propto \omega^{1+p}$ according to Equation (2), with a power exponent $n$ value higher than 2. As an example, a simulation was carried out based on the fitting parameters values of the higher frequency vibration mode of Figure 3. Figure 4 shows the contribution of this vibration mode to dielectric losses as a function of frequency, as well as the derivative of the function $\log \varepsilon''$ $(\log f)$ which represents the slope $p$. From frequency 1.77 THz ($\log f = 12.25$) up to peak frequency 13.5 THz ($\log f = 13.13$), almost one order of magnitude, the slope gradually increases and take values in the range between 1 and 6.3. Therefore, in the extended frequency region, just below $\omega_o$, the power exponent of AC conductivity can take values higher than 2 in some cases, depending on the correlation of DHO parameters values, $\gamma$ and $A$. This finding could explain the experimental data in glycerol [31] where the frequency dependence of $\varepsilon''$ follows a power law $\omega^3$ at 400–900 GHz leading to $\sigma'_V \propto \omega^4$.

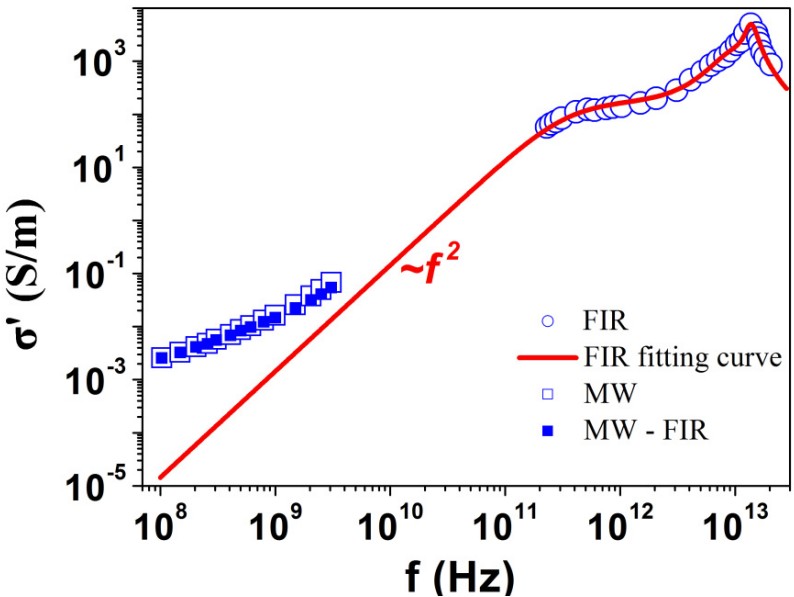

**Figure 3.** Real part of AC conductivity as a function of frequency of a lithium silicate glass. The blue open symbols represent experimental data points of FIR (circles) and MW (squares) frequencies (Data were taken from literature [29]). The blue solid squares represent MW data after the subtraction of FIR contribution. The red line is the best fitting of a sum of three terms $\varepsilon_o \omega_i \text{Im}[DHO]_i$. The fitting parameters' values are: $\gamma_1/2\pi = 1.12 \times 10^{13}$ (Hz), $A_1/(2\pi)^2 = 3.20 \times 10^{25}$ (Hz$^2$), $\omega_{o1}/2\pi = 1.94 \times 10^{12}$ (Hz), $\gamma_2/2\pi = 9.71 \times 10^{12}$ (Hz), $A_2/(2\pi)^2 = 1.98 \times 10^{26}$ (Hz$^2$), $\omega_{o2}/2\pi = 9.89 \times 10^{12}$ (Hz), $\gamma_3/2\pi = 3.79 \times 10^{12}$ (Hz), $A_3/(2\pi)^2 = 2.84 \times 10^{26}$ (Hz$^2$) and $\omega_{o3}/2\pi = 1.37 \times 10^{13}$ (Hz).

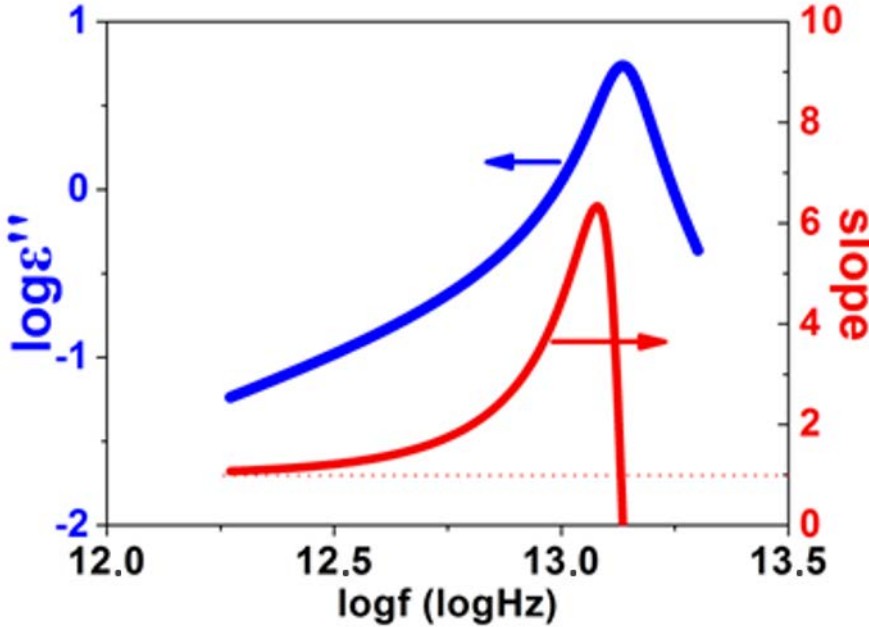

**Figure 4.** The blue line (left axis) shows the frequency dependence of the imaginary part of complex dielectric constant according Equation (9) with parameter values $\gamma/2\pi = 3.79 \times 10^{12}$ (Hz), $A/(2\pi)^2 = 2.84 \times 10^{26}$ (Hz$^2$) and $\omega_o/2\pi = 1.37 \times 10^{13}$ (Hz) (higher frequencies vibration mode of Figure 3). The red line (right axis) shows the corresponding slope of $\log \varepsilon''$ ($\log f$) curve.

## 4. Conclusions

In the present work, the AC conductivity response at the high frequency regime is investigated. The existence of vibrational modes, in the THz region, has a decisive influence in the power exponent of AC conductivity values in the lower frequencies of the FIR range. The vibrational modes contribution, as described from the DHO equation, results in the gradual increase of the power exponent which leads to superlinear values, while at the lower frequencies of FIR, well below the resonance frequency $\omega_o$, could approach values up to 2. In these cases, the power exponent values depend on the dynamic characteristics of the vibration modes and especially on the correlation of DHO parameter values. For a relative extended frequency region, just below resonance frequency $\omega_o$, values of the power exponent of AC conductivity higher than 2 cannot be excluded.

**Funding:** This research received no external funding.

**Institutional Review Board Statement:** Not applicable.

**Informed Consent Statement:** Not applicable.

**Data Availability Statement:** Data supporting this article are available from the corresponding author upon reasonable request.

**Conflicts of Interest:** The author declares no conflict of interest.

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
