# Peer review of "Exploring the High Frequencies AC Conductivity Response in Disordered Materials by Using the Damped Harmonic Oscillator"

_jcs, doi:10.3390/jcs6070200_

Round 1
Reviewer 1 Report
Terahertz spectroscopy is continually gaining ground and many interesting properties of materials have been discovered and/or explained using this technique. This manuscript expands on a model, presented in a previous publication, based on the universal dynamic response (UDR) to fit AC conductivity data in the gigahertz (GHz) to terahertz (THz) region. The intent is to better explain the behavior of disordered and composite materials in that frequency range.
What results is a facile yet effective summation of well known DC conductivity, Havriliak-Nagami (HN) and damped harmonic oscillator (DHO) equations to obtain a more comprehensive description of relaxtion and vibrational mechanisms in real disordered materials. This is shown by providing a much better fit to AC conductivity vs. frequency data for poly(p-phenylenediamine), especially in the GHz to low THz region. The derivation of the AC conductivity model explains the range of values for "n", the exponent of angular frequency in the UDR, at least to the low THz range, as shown in the simulated curves of Fig. 1.
The paper is well written with clear and well described equations, images and table. It is interesting that by simply summing appropriate numbers of HN relaxations and DHO vibrational modes, such good fits can be acheived for over 13 decades of real spectroscopy data. However, I do have a few minor comments focused on what is shown in Figure 3:
1. Why was the sub-THz region not fit with the full AC conductivity model presented? If it was, what would its effect be on having 3 DHO's be the best fit for the THz region? The HN model does have an influence on the increasing slope extending from the GHz, so I think a fit using all of Eq. 9 would be good to show applicability to an inorganic glassy system.
Author Response
Response to Reviewer 1
Dear Reviewer,
I would like to thank you very much for your valuable comments concerning our article submitted to “Journal of Composites Science”.
Comment
However, I do have a few minor comments focused on what is shown in Figure 3:
- Why was the sub-THz region not fit with the full AC conductivity model presented? If it was, what would its effect be on having 3 DHO's be the best fit for the THz region? The HN model does have an influence on the increasing slope extending from the GHz, so I think a fit using all of Eq. 9 would be good to show applicability to an inorganic glassy system.
Response
Indeed, would be good to show the applicability of Εq. 9 to an inorganic glassy system. However, since in Fig. 3 there is no experimental data below 100 MHz as also in the region 3-300 GHz where the fast relaxation manifests, the full fitting process of Eq. 9 is questionable. There are many combinations of the HN parameter values, in the respective frequency regions (below 100 MHz and 3 -300 GHz), that give very similar fitting results. In Fig. 3, what emerges is that in glassy systems the superlinear value of power exponent n near MHz-GHz region is not due to vibration modes contribution but to the fast relaxation.
Reviewer 2 Report
Referee Report
on paper “ Exploring the high frequencies AC conductivity response in disordered materials by using the damped harmonic oscillator “ (jcs-1814462) by author Christos Tsonos submitted to Journal of Composites Science
This is interesting theoretical paper. It reports the definitions of electromagnetic theory as well as the Havriliak-Negami equation and the damped harmonic oscillator equation, which are widely used for the description of dielectric relaxation mechanisms and vibration modes in THz frequencies region. The obtained theoretical results are reliable without any doubts. However, I have some questions and additions. I would like to note a few points to improve the paper before it can be published:
1. The authors should give in 1. Introduction the specific examples of materials promising for microwave applications:
(1). A.V. Trukhanov, V.O. Turchenko, I.A. Bobrikov, S.V. Trukhanov, I.S. Kazakevich, A.M. Balagurov, Crystal structure and magnetic properties of the BaFe12−xAlxO19 (x=0.1–1.2) solid solutions, J. Magn. Magn. Mater. 393 (2015) 253-259. https://doi.org/10.1016/j.jmmm.2015.05.076.
(2). D.A. Vinnik, V.E. Zhivulin, D.P. Sherstyuk, A.Yu. Starikov, P.A. Zuzina, S.A. Gudkova, D.A. Zherebtsov, K.N. Rozanov, S.V. Trukhanov, K.A. Astapovich, V.A. Turchenko, A.S.B. Sombra, D. Zhou, R.B. Jotania, C. Singh, A.V. Trukhanov, Electromagnetic properties of zinc-nickel ferrites in frequency range of 0.05-10 GHz, Mater. Today Chem. 20 (2021) 100460. https://doi.org/10.1016/j.mtchem.2021.100460.
2. It is well known that the combination of different compounds which have excellent microwave properties leads to new composite materials which have earned great technological interest in recent years. The addition of a second phase can significantly improve the electronic properties of the resulting composite material:
(3). M.A. Almessiere, A.V. Trukhanov, Y. Slimani, K.Y. You, S.V. Trukhanov, E.L. Trukhanova, F. Esa, A. Sadaqat, K. Chaudhary, M. Zdorovets, A. Baykal, Correlation between composition and electrodynamics properties in nanocomposites based on hard/soft ferrimagnetics with strong exchange coupling, Nanomaterials 9 (2019) 202. https://doi.org/10.3390/nano9020202.
(4). M.A. Almessiere, Y. Slimani, N.A. Algarou, M.G. Vakhitov, D.S. Klygach, A. Baykal, T.I. Zubar, S.V. Trukhanov, A.V. Trukhanov, H. Attia, M. Sertkol, I.A. Auwal, Tuning the structure, magnetic and high frequency properties of Sc-doped Sr0.5Ba0.5ScxFe12-xO19/NiFe2O4 hard/soft nanocomposites, Adv. Electr. Mater. 8 (2022) 2101124. https://doi.org/10.1002/aelm.202101124.
This issue should be mentioned and discussed in 1. Introduction.
3. The proposed 4 papers should be inserted in References.
The paper should be sent to me for the second analysis after the moderate revisions.
Author Response
Response to Reviewer 2
Dear Reviewer,
I would like to thank you very much for your valuable comments and remarks concerning our article submitted to “Journal of Composites Science”. Following your suggestions, some changes were made to the text.
Comments
I would like to note a few points to improve the paper before it can be published:
- The authors should give in 1. Introduction the specific examples of materials promising for microwave applications:
(1). A.V. Trukhanov, V.O. Turchenko, I.A. Bobrikov, S.V. Trukhanov, I.S. Kazakevich, A.M. Balagurov, Crystal structure and magnetic properties of the BaFe12−xAlxO19 (x=0.1–1.2) solid solutions, J. Magn. Magn. Mater. 393 (2015) 253-259. https://doi.org/10.1016/j.jmmm.2015.05.076.
(2). D.A. Vinnik, V.E. Zhivulin, D.P. Sherstyuk, A.Yu. Starikov, P.A. Zuzina, S.A. Gudkova, D.A. Zherebtsov, K.N. Rozanov, S.V. Trukhanov, K.A. Astapovich, V.A. Turchenko, A.S.B. Sombra, D. Zhou, R.B. Jotania, C. Singh, A.V. Trukhanov, Electromagnetic properties of zinc-nickel ferrites in frequency range of 0.05-10 GHz, Mater. Today Chem. 20 (2021) 100460. https://doi.org/10.1016/j.mtchem.2021.100460.
- It is well known that the combination of different compounds which have excellent microwave properties leads to new composite materials which have earned great technological interest in recent years. The addition of a second phase can significantly improve the electronic properties of the resulting composite material:
(3). M.A. Almessiere, A.V. Trukhanov, Y. Slimani, K.Y. You, S.V. Trukhanov, E.L. Trukhanova, F. Esa, A. Sadaqat, K. Chaudhary, M. Zdorovets, A. Baykal, Correlation between composition and electrodynamics properties in nanocomposites based on hard/soft ferrimagnetics with strong exchange coupling, Nanomaterials 9 (2019) 202. https://doi.org/10.3390/nano9020202.
(4). M.A. Almessiere, Y. Slimani, N.A. Algarou, M.G. Vakhitov, D.S. Klygach, A. Baykal, T.I. Zubar, S.V. Trukhanov, A.V. Trukhanov, H. Attia, M. Sertkol, I.A. Auwal, Tuning the structure, magnetic and high frequency properties of Sc-doped Sr0.5Ba0.5ScxFe12-xO19/NiFe2O4 hard/soft nanocomposites, Adv. Electr. Mater. 8 (2022) 2101124. https://doi.org/10.1002/aelm.202101124.
This issue should be mentioned and discussed in 1. Introduction.
- The proposed 4 papers should be inserted in References.
Response
In line 73 was added (red color)
On the other hand, the higher frequency range is an interesting region for the study of promising materials for microwave applications [32, 33]. The combination of different compounds which have excellent microwave properties leads to new composite materials which have earned great technological interest in recent years, while the addition of a second phase can significantly improve the electronic properties of the resulting composite material [34, 35].
In line 434, the below four references were added
- A.V. Trukhanov, V.O. Turchenko, I.A. Bobrikov, S.V. Trukhanov, I.S. Kazakevich, A.M. Balagurov, Crystal structure and magnetic properties of the BaFe12−xAlxO19 (x=0.1–1.2) solid solutions, J. Magn. Magn. Mater. 393 (2015) 253-259.
- D.A. Vinnik, V.E. Zhivulin, D.P. Sherstyuk, A.Yu. Starikov, P.A. Zuzina, S.A. Gudkova, D.A. Zherebtsov, K.N. Rozanov, S.V. Trukhanov, K.A. Astapovich, V.A. Turchenko, A.S.B. Sombra, D. Zhou, R.B. Jotania, C. Singh, A.V. Trukhanov, Electromagnetic properties of zinc-nickel ferrites in frequency range of 0.05-10 GHz, Mater. Today Chem. 20 (2021) 100460.
- M.A. Almessiere, A.V. Trukhanov, Y. Slimani, K.Y. You, S.V. Trukhanov, E.L. Trukhanova, F. Esa, A. Sadaqat, K. Chaudhary, M. Zdorovets, A. Baykal, Correlation between composition and electrodynamics properties in nanocomposites based on hard/soft ferrimagnetics with strong exchange coupling, Nanomaterials 9 (2019) 202.
- M.A. Almessiere, Y. Slimani, N.A. Algarou, M.G. Vakhitov, D.S. Klygach, A. Baykal, T.I. Zubar, S.V. Trukhanov, A.V. Trukhanov, H. Attia, M. Sertkol, I.A. Auwal, Tuning the structure, magnetic and high frequency properties of Sc-doped Sr0.5Ba0.5ScxFe12-xO19/NiFe2O4 hard/soft nanocomposites, Adv. Electr. Mater. 8 (2022) 2101124.
The numbering of the references after Ref. [32] of the initial version was changed accordingly.
Round 2
Reviewer 2 Report
Referee Report
on paper “ Exploring the high frequencies AC conductivity response in disordered materials by using the damped harmonic oscillator “ (jcs-1814462-v2) by author Christos Tsonos submitted to Journal of Composites Science
This paper has been well corrected and it should be published urgently.
